# Viticulture in the Laetanian Region (Spain) during the Roman Period: Predictive Modelling and Geomatic Analysis

**Lisa Stubert [1], Antoni Martín i Oliveras [2], Michael Märker [3]** 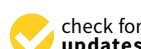**, Harald Schernthanner [4]** **and Sebastian Vogel [5],***

[1]  Technologiestiftung Berlin, 10825 Berlin, Germany; lisastubert@yahoo.de
[2]  Department of History and Archaeology, University of Barcelona, 08028 Barcelona, Spain; amartinioliveras@ub.edu
[3]  Department of Earth and Environmental Sciences, University of Pavia, 27100 Pavia, Italy; michael.maerker@unipv.it
[4]  Institute of Geosciences, University of Potsdam, 14476 Potsdam, Germany; hschernt@uni-potsdam.de
[5]  Engineering for Crop Production, Leibniz Institute for Agricultural Engineering and Bioeconomy, 14469  Potsdam, Germany
*  Correspondence: svogel@atb-potsdam.de

**Abstract:** Geographic information system (GIS)-based predictive modelling is widely used in archaeology to identify suitable zones for ancient settlement locations and determine underlying factors of their distribution. In this study, we developed predictive models on Roman viticulture in the Laetanian Region (*Hispania Citerior-Tarraconensis*), using the location of 82 ancient wine-pressing facilities or *torcularia* as response variables and 15 topographical and 6 socio-economic cost distance datasets as predictor variables. Several predictor variable subsets were selected either by expert knowledge of similar studies or by using a semi-automatization algorithm based on statistical distribution metrics of the input data. The latter aims at simplifying modelling and minimizing the necessity of a priori knowledge. Both approaches predicted the distribution of archeological sites sufficiently well. However, the best prediction performance was obtained by an expert knowledge model utilizing a predictor variable combination based on recommendations on viticulture by *Lucius Junius Moderatus Columella*, the prominent ancient Roman agronomist. The results indicate that the accessibility of a location and its connectivity to trade routes and distribution centres, determined by terrain steepness, was decisive for the settlement of viticultural facilities. With the knowledge gained, the ancient cultivated area and number of wine-pressing facilities needed for processing the vineyard yields were extrapolated for the entire study region.

**Keywords:** geoarcheology; GIS automatization; Python; cost distance analysis; geostatistics; wine presses

## 1. Introduction

The application of geographical information systems (GIS) has considerably increased in the last decades [1] and can be used for a wide variety of subjects and research questions. In the 1990s, one particular scientific field experienced a 'boom' in GIS applications: archaeology [2]. One issue that has always been important in the application of archaeological GIS is predictive modelling. Their first occurrence dates back to the 1970s, but their development especially increased with the widespread popularity of GIS [2]. The aim of applying a predictive model is to envisage the potential of undiscovered archaeological site locations based on observed patterns in the environment and/or

assumptions about human behaviour [3,4]. The basis of such models is that the spatial distribution of cultural remains, which are represented in archaeological sites, is the result of human decision-making activities within the framework of the surrounding environment. Predictive models have many advantages: They are cost-effective and useful for the identification, protection, and management of increasingly threatened cultural resources. Most contemporary predictive modelling approaches combine information from multi-thematic location characteristics relating to past environmental and/or cultural conditions [5]. These are the predictor variables. With them, the suitability of areas for the occurrence of archaeological sites, i.e., the response variable, can be determined. The more frequent use of predictive modelling can, therefore, certainly also be attributed to the increasing availability of data that can be used as predictor variables. Remote sensing can complement conventional observation and data acquisition even in areas that are difficult to capture by other methods. In particular, high-resolution digital elevation models (DEM) can be used to derive important landscape parameters and, therefore, variables for modelling.

Freimark [6] stated that this technological progress has created a gap between those who learned to use systems like GIS and those who did not. For predictive modelling in particular, a major problem can arise here, because its approach holds many potential sources of error. This is for example the use of inappropriate statistical techniques or emphasis on less appropriate factors [2]. As a consequence, competences from different fields are required, namely in data preparation and model execution as well as the necessary expert knowledge on the research subject and the study area. Furthermore, the final model output is often poorly presented and, thus, not easy to interpret by the archaeologists [6]. Therefore it is also important to select a suitable visualization with as much added value as possible, in order to make the results available to the scientists who are evaluating them.

In recent years, a number of research studies have dealt with the complex interrelationships between the development of ancient societies, economy and landscape characteristics. A special and very extensive subject area is the Roman period and its agriculture, to which many studies have contributed by means of predictive models and other types of geospatial analyses. Some of these works refer to the regions close to the origin of the Roman Empire on today's Italian peninsula [7–9], others to the conquered provinces of today's Western and Middle Europe [2,10–12].

In the present paper, for the first time, predictive modelling was performed in the spatial context of the Laetanian Region, located at the present Catalan coast in Northwest Spain, for a dataset of archeological sites related to viticulture during the Roman period (around 100 BC and 400 AD). A large set of possible predictor variables was prepared for the modelling, including terrain characteristics derived from a high-resolution DEM and several socio-economic location characteristics which were considered by other studies to be potentially important driving forces. The work is based on the hypothesis that the distribution of Roman viticultural properties or *fundus* and winemaking facilities, largely depended on these environmental and socio-economic characteristics. These characteristics are used to predict suitability areas for settlement and viticulture and to perform an estimation about the vineyard's cultivated area and number of wine-pressing facilities or *torcularia* that potentially existed in Roman times. The results serve to deepen our understanding of the distribution of archaeological sites and to draw conclusions about the factors that were decisive for the development of ancient rural settlements and its evolution over the time.

A further objective of this paper is to semi-automate the process of predictive modelling by developing a script in the programming language Python. Since not all predictor variables contribute to a significant improvement of the model and in order to obtain a trustworthy model showing a good performance, two different approaches for variable selection were tested. The predictor variables were either selected by expert knowledge of two similar studies [8,10] or by using a semi-automatization algorithm based on measures of statistical dispersion. This latter method is intended to close the aforementioned gap between the geospatial analyses experts and the archaeologists. *a priori* knowledge less important. Finally, the results will be presented in a visually appealing way for further analysis by the archaeologists in an interactive web map.

## 2. Materials and Methods

### 2.1. Research Area

The Laetanian Region, although not clearly defined, is an Iberian ancient territory located in the Northwest of *Hispania Citerior-Tarraconensis* Roman province on the present-day Catalan coast of Spain (see Figure 1). It includes several urban centres that already existed in Roman times including *Blandae* (today Blanes), *Iluro* (today Mataró), *Baetulo* (today Badalona) and the colony of *Barcino* (today Barcelona). The study area is bordered to the south and southeast by the Mediterranean Sea and has a flat coastal strip of varying width, which merges into a littoral mountain range towards the northwest. In the hinterland stretches the extensive Vallesian Plain situated between the *Arnum* River (today River Tordera) and the *Baetulo* River (today River Besós), and the smaller Barcelona's Plain situated between the latter and the *Rubricatum* River (today River Llobregat; [13]). These main rivers supplied fluvial sediments which were redistributed southwards by longshore marine currents and shaped the coastal and wetland plains of the present Maresme, Barcelonès and Baix Llobregat shires [14,15]. To the north, the Vallesian Plain is bordered by another pre-littoral mountainous range. On the northwestern edge of this plain, the Flavian *municipium* of *Egara* is situated. To the south, the thin coastal fringe ends in the foothills of the Garraf massif [13].

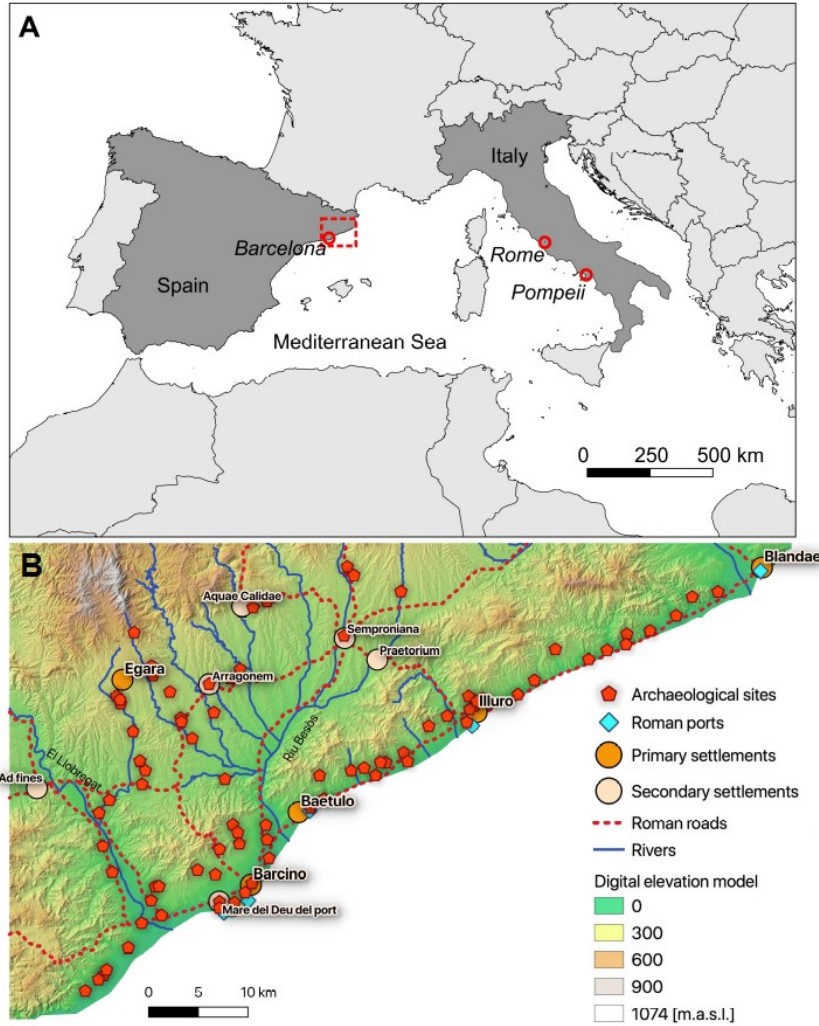

**Figure 1.** (**A**) Map of Southern Europe with the location of the research area (red rectangle) and (**B**) digital elevation model of the Laetanian Region with the locations of archaeological sites and socio-economic points of interest.

## 2.2. Romanization and Viticulture

The present Catalan coast was marked by great changes in the 2nd and 1st centuries BC. In those times, the Iberian culture faced the presence of the Romans who had arrived at the Iberian Peninsula one century earlier during the 2nd Punic War which had an impact on its economic, social and cultural structure [14]. The Romanization process transformed the entire region including: (i) the founding of Roman settlements or the reshaping of existing ones, (ii) the building of a vast road network and ports, and (iii) the adoption of new agricultural exploitation forms, as investigated by several authors [13,15–17]. The Roman economy was an agrarian regime that produced mainly cereal, wine and olive oil combined with livestock farming. In the Laetanian Region, an intensive viticulture was practiced as a widespread phenomenon having huge economic implications on the organization of the territory between the 1st century BC and 3rd century AD. Viticulture was integrated into the concept of the Roman "*villa* system". These *villae* were progressively specialized as winegrowing farms, often containing a sector of private spaces (*pars urbana*) for the *dominus* (owner) and his family or for the *villicus* (foreman), either tenant or servant, and other areas called *pars rustica* and *pars fructuaria* used for agricultural production. For winemaking processes, these latter areas included several presses (*torculares*), must collection depots (*lacus*) and storage facilities (*cellae vinariae*).

In the Laetanian Region, the wine was not only produced for regional consumption, but was also traded interregionally and overseas. Evidence of this is the fact that some of the *villae* had kilns and pottery workshops (*figlinae*) for *amphorae* production included or nearby. These *amphorae* were used as containers for nautical transport of the wine and its derivates, such as vinegar (*acetum*), spiced honey wine (*mulsum*), grape must syrups (*defruntum*, *sapa*, *caroenum*), etc. [13,17].

## 2.3. Dataset

### 2.3.1. Archaeological Dataset

The archaeological dataset of ancient wine pressing facilities and pottery kilns from Laetanian Region have been collected by Martín i Oliveras [17] according to Revilla [18] and Tremoleda [19]. They are point geometries that contain information on 82 sites where wine presses were found or assumed. It also contains the coordinates and data of important primary and secondary settlements that acted as nodes for the trade and distribution of goods. In addition to the coordinates, the table contains the number of wine presses or kilns (estimated and actually found) and the starting and ending chronology of usage for the corresponding facility. In general, the buildings for viticultural production were found close to the residential areas but they could also be a little farther away or even set apart [13,17]. This is the reason why the modelling of the sites does not directly point to the *villae*, in the sense of their residential area. In contrast, in the following, the location of sites refers basically to winegrowing farms with real or estimated pressing facilities. Hence, the locations of the presses were used as the response variable in the predictive model. It is hypothesized that the known archaeological sites represent a random subsample of all site that have existed in the past. Thus, they essentially reflect the distribution of all wine-pressing facilities for each chronological period. Their suitability is to be determined by the respective topographic and socioeconomic location characteristics of the viticultural facilities. In order to take into account all the surrounding land cultivated by these agrarian settlements and to avoid errors due to possible inaccuracies in the localization process of the sites, the point geometry was transformed into buffer polygons covering a wider area around the actual archaeological site. This procedure is based on the assumption that the entire vineyard estate of each wine-pressing facility is of particular importance for its detailed characterization [8]. Therefore, the location characteristics inside the buffer area were averaged. To determine the effect of different buffer sizes on the modeling results, buffers were tested having diameters of 50 m ($\cong962$ m$^2 \cong \frac{1}{2}$ *iugera*), 100 m ($\cong7850$ m$^2 \cong3$ *iugera*) and 250 m ($\cong49,062$ m$^2 \cong20$ *iugera*). An agronomical modulation of 100 or 200 *iugera* is the most common Roman standard parcel which we recently used for microeconomic calculation studies of vineyard's crop and winemaking facilities' processing yields [20].

### 2.3.2. Topographic Dataset and Socio-Economic Dataset

The basis of the predictor variables is a gridded DEM representing the terrain surface as a regular lattice of point elevations [21]. The DEM used for this study was downloaded from the *Institut Cartogràfic i Geològic de Catalunya*. It contains the orthometric height in a 5 × 5 m resolution and an estimated altimetric accuracy of 0.90 m mean square error [22]. To keep the data size small and calculation times low, the DEM was resampled to a resolution of 10 × 10 m. In the DEM, the Mediterranean Sea was converted to a no-data value of 9999 as it will not be relevant for the predictive model.

The available DEM can only reflect today's topographical situation of the study area. However, the concept of uniformitarianism in geology [23,24] assumes that the shape of the Earth is the result of currently existing processes that have operated in the same way in the geological past, even though with different speed and intensity. Hence, processes that are identified today also occurred on former landscapes, i.e., the present is a key to the past. Therefore, it is valid to use the present day topography as a representation of the past conditions. However, the coastline particularly near Barcelona has propagated into the sea due to centuries of fluvial sedimentation [14,15,25]. In order to get closer to the former topography, the present-day DEM was clipped with the ancient coastline based on reconstructions of the *Institut Cartogràfic i Geològic de Catalunya* [26] and Riera et al. [25]. Afterwards, the elevation above sea level (a.s.l.) was recalculated on this new base level by modelling the vertical distance to the ancient coastline. This resulting DEM can be considered as an approximation to the topography in Roman times.

The elevation is not only a topographical factor itself, but all other topographical variables were deduced from it. In addition, the cost distances used as socio-economic location characteristics were also calculated from the slope, a derivate of the elevation. For this study, the primary attributes slope, aspect, curvature, profile curvature and additionally 10 secondary attributes were calculated and considered as potential predictor variables (Table 1). Only metric variables were considered because the statistical parameters for the modelling can only be calculated for variables that are cardinally scaled. The calculation of the topographic variables was carried out using the open-source software SAGA GIS [27].

**Table 1.** Overview of all terrain characteristics derived from the ancient digital elevation model (DEM) using SAGA geographic information system (GIS) terrain modules. If a specified algorithm was used, it is given in the description.

| Variable | Description |
| --- | --- |
| Slope | The rate of change of elevation in the direction of steepest descent [21]. Method used: Zevenbergen and Thorne [28]. |
| Aspect | Orientation of the line of steepest descent [21]. Method used: Zevenbergen and Thorne [28]. |
| Curvature | Measures the change of slope as a degree of concavity and convexity. This determines flow velocity and erosion rate. Method used: Zevenbergen and Thorne [28]. |
| Profile curvature | Measures the rate of change of slope only along a flow line. This indicates acceleration or deceleration of flow [21]. Method used: Zevenbergen and Thorne [28]. |
| LS-factor | The S-factor is the slope steepness, the L-factor the slope length. In relation they determine soil erosion [29]. Method used: Moore [30]. |
| Terrain surface texture (TS texture) | Measures the 'grain' of terrain. Each raster cell value represents the relative frequency of pits and peaks within a radius of ten cells [31]. |
| Topographic position index (TPI) | Comparison of the elevation of each cell to the mean elevation of a specified neighborhood around that cell [32]. |
| Terrain ruggedness index (TRI) | Measure of topographic heterogeneity. Each cell is the sum change in elevation between itself and its eight neighboring cells [33]. |
| Topographic wetness index (TWI) | Describes the spatial distribution and extent of zones of water saturation and therefore the runoff generation [33]. |
| Skyview | Fraction of the sky that can be seen from the soil surface, given by an index between 0 (plain or peaks) and 1 (completely obstructed). It indirectly indicates the exposure to the wind [21,34]. |
| Direct insolation | Intensity of potential direct solar irradiation, assuming clear-sky conditions. It is affected by topographic shading (shading by nearby hills) [21,34]. |
| Diffuse insolation | Intensity of potential diffuse solar irradiation, assuming clear-sky conditions. It increases with decreasing altitudes, because of aerosol, water droplets and water vapor scattering the solar radiation [34]. |
| Diurnal anisotropic heating (DAH) | Combination of the effects of slope and aspect. Indicates temperature and topographic solar radiation at the soil surface [34,35]. |
| Vertical distance to channel network | Elevation of a cell that was calculated from the difference between the original elevation and the elevation of the closest channel. The channels were calculated from the catchment area. |

In addition to the 15 topographical characteristics, 6 socio-economic factors were included in the model. As part of the physical environment, the connectivity and transport infrastructure of the

Laetanian Region was decisive for an efficient operational flow of the production and trade system as it determined the movement of goods [13]. The following socio-economic factors were considered in the modelling:

i. Cities:

The point locations of the ancient Roman cities (hereafter referred to as primary settlements) *Barcino*, *Baetulo*, *Blandae*, *Iluro* and *Egara* and the smaller, secondary settlements such as *Ad fines* (today Martorell), *Arragonem* (today Sabadell), *Aquae Calidae* (today Caldes de Montbui), *Semproniana* (today Granollers) and *Praetorium* (today Llinars del Vallès) were used [36]. It is assumed that settlements interacted as communication nodes. Thus, they were important consumption and redistribution centres for the local or regional wine producers and dealers [37].

ii. Ports:

The Roman ports had the function of concentration points and trade redistribution nodes. They stored goods, like the winegrowing merchandise, and organized trade route expeditions [38]. Some settlements benefited from the construction of port infrastructures that connected them to Rome and many other cities [16]. In the Laetanian Region, five hypothetical Roman seaports have been assumed, related to and located near the main coastal settlements or in strategic areas of its long seashore [36,38,39]: (i) one port each at *Blandae*, *Iluro* and *Baetulo municipia*, (ii) two ports related to *Barcino colonia* (one situated close to the *Rubricatum* River, the other one at the coast in front of the ancient Roman city), and (iii) one fluvial dock documented inland at *Ad fines*. It is also disputed whether the *Rubricatum* area harbour was at Mare de Déu del Port Roman settlement located in the southern foothills of the Montjuïc mountain or in another place near the river mouth [38,40]. As the hypothetical harbour areas were located at or near the main settlements, for the predictive modelling, it was decided not to create an independent variable for the ports, to avoid correlations between predictor variables.

iii. Anchorage areas:

Additionally to the ports, different anchorage areas have been documented. They were also used for local transport and cargo operations. Small boats supplied wine containers (*amphorae*, wineskins and wooden barrels) from the beaches to larger ships [38,40]. As anchorage areas covered the entire seashore of the Laetanian region, this variable is well represented by the ancient coastline. To avoid the use of duplicate datasets, the anchorage areas were excluded from the modelling.

iv. Waterways:

For the waterways, which are considered an important transport route for goods from the inland to the coast [16,40], the Overpass API (Application Programming Interface) was used to extract rivers and smaller streams from OpenStreetMap. However, the dataset generated contains a very high density of present-day waterways. They most likely have changed since Roman times and some waterways, such as artificial canals, did not even exist at that time. In order to keep the error source as small as possible, the dataset was therefore carefully filtered to the main natural local streams.

v. Roads:

Another possibility for moving goods was the transport by terrestrial vehicles on roads. The Roman road dataset was digitized by de Soto [16] from detailed archaeological studies of different authors on transport networks in Roman *Hispaniae*. The resulting digitized roads were differentiated into primary and secondary roads, depending on their role in the communication of the territories. Primary roads have been collected in itineraries and ancient sources due to their supra-regional importance. Secondary roads are perpendicular or radial branches that connected the coastal territories with the inland valleys, plains and mountainous areas. A curious example in the Laetanian Region is the splitting of the Via Augusta in two arms at *Aquis Voconis* (today Caldes de Malavella). One runs inland through the Vallesian Plain and the other one runs along the coastal fringe as far as *Ad Fines* where they joined again to run along the coast connecting the northeast of *Hispania Citerior Tarraconensis* with the southern province of *Hispania Ulterior Baetica*. Decisive for their classification is their morphology and their position within the transport network. For the present study, it is assumed that the proximity

to the road was generally an advantage for the settlement of wine-pressing facilities. All ancient roads were combined into a single socio-economic factor.

In total, five socio-economic factors related to transport and distribution of goods have been chosen as predictor variables in the modelling: primary settlements, secondary settlements, the coastline, roads network, main rivers and local streams. To convert these point (settlements) or line (coastline, roads, rivers) vector objects into a continuous dataset of metric values for the entire study area, distance rasters were calculated. Given the pronounced topography of the Laetanian Region, instead of the Euclidean distance, terrain-sensitive cost distance calculations were used that are based on a cost surface, (sometimes also referred to as friction surface or effort model). There are a lot of different functions proposed by several authors to obtain a cost surface. Nearly all of them are based on the slope of the terrain, often combined with other factors and adapted to motion of pedestrians, horse riders and wheeled transport carriages pulled by oxen, named *plaustra* [41,42].

For this study, an approach was used that showed good results in a study about modelling the routes of Roman roads in the northwestern Iberian Peninsula stated by Parcero-Oubiña et al. [43]. They used a polynomial slope-dependent function proposed by Llobera and Sluckin [44] that expresses the energy expenditure increase needed to climb steep slopes instead of walking on flat land. Additionally, extra costs were added to raster cells with elevations ≥1050 m a.s.l. to avoid high altitudes as well as to raster cells that intersect the river beds (here, river line geometries were buffered by 20 m). Furthermore, the function was fitted to model the use of wheeled vehicles. Fonte [45] propose extra costs for slopes >8% to 16% depending on terrain characteristics. In this study, the minimum slope for extra costs was set to 16%.

The generated cost surface was then used to calculate cost distance rasters for each socio-economic location raster. Since this technique is applicable to cultural as well as environmental variables, another variable was created, which is actually based on the topography. This was inspired by another study: Vaughn and Crawford [46] identified areas of high archaeological potential related to agriculture of the Maya in Northwest Belize. They assumed that the Maya have valued large amounts of arable land within a close proximity to their settlements. Therefore, they calculated the distance to areas with slopes that were known as suitable and included it as a predictor variable in their modelling. This approach was also applied in the present study. Important conditions for present-day vineyards are favorable slopes and aspects. Slopes between 5% and 15% provide sufficient air circulation to reduce the chance of infections with fungal or bacterial diseases. Moreover, erosion and nutrient loss is moderate [47]. In the case of aspect, southern, southwestern and southeastern slopes are preferred [48]. Based on the hypothesis that these conditions were similar for ancient viticulture, favorable hillsides were identified in the study area. The cost distance to favorable hillsides is assumed to be useful for the modelling in such cases where the real ancient vineyard was not in direct vicinity to an agrarian or urban settlement and, thus, cannot be covered by the buffer area around the wine pressing facility. Figure 2 shows all socio-economic vector datasets, the cost surface and the resulting cost distance rasters.

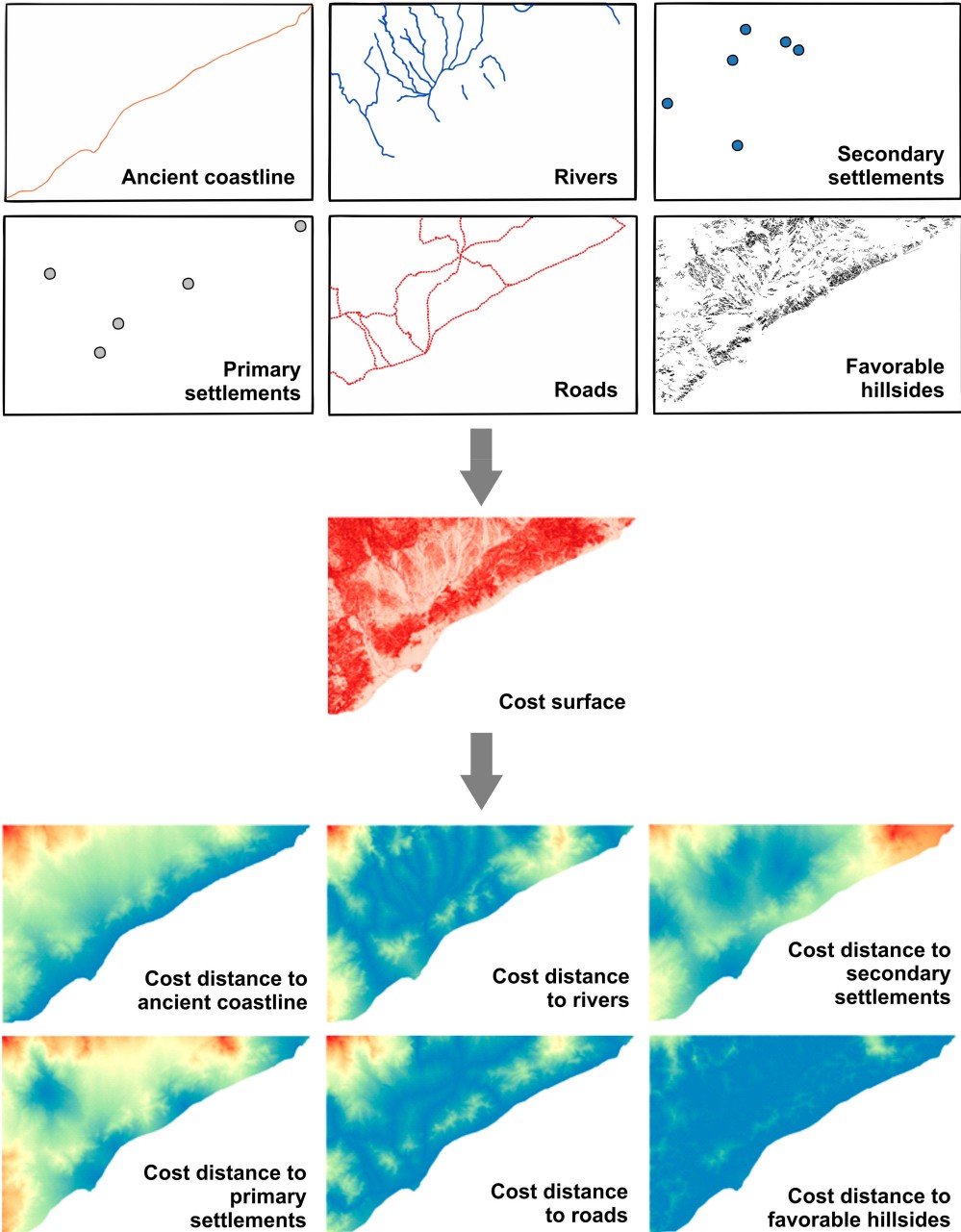

**Figure 2.** Overview of all socio-economic factors and cost distance rasters used for the modelling.

*2.4. Predictive Modelling*

2.4.1. Modelling Modules and Automatization Procedure

The general modelling methodology described in the following was developed by Vogel [8]. However, in the present study, it was largely refined. One major objective is to semi-automate the modelling process since not all topographic and socio-economic characteristics at hand contribute to an improvement of the model performance. Consequently, two different modelling approaches were applied and compared. At first, the modelling was first carried out using predictor variables that are selected on the basis of expert knowledge and the results of previous research studies. In a second approach, the predictive models were calculated with subsets of variables that were automatically selected based on measures of statistical dispersion. In the present study, the programming language Python [49] was used. The developed script is based on several modules, which are executed by

sub-scripts. Each module combines partial steps of the methodology to a logical unit and is explained in the order of execution:

i. Extraction of average terrain characteristics of buffered sites:

The first module calculates the average of all cells from a raster layer within the buffered archeological site. Each site is overlaid with one raster and all raster cells intersecting with the buffer polygon are collected and the mean value is calculated. This calculation is iterated over all buffer polygons and all rasters of predictor variables.

ii. Generation of box plot statistics of datasets:

In order to make the individual variables and their box plots comparable, the datasets were normalized. In the next step the box plot statistics of the normalized dataset of each raster is obtained by splitting the dataset by its quantiles and analyzing it by its structure [50]. Each dataset can be graphically represented by its box and whiskers. The box contains 50% of the data points and, therefore, all those within the so-called lower and upper quartile (25th and 75th percentile or $x_{0.25}$ and $x_{0.75}$). The range of these data (the difference between $x_{0.25}$ and $x_{0.75}$) is the interquartile range (IQR). As we work with normalized data, we refer to the normalized IQR ($IQR_{norm}$) of the original dataset:

$$IQR_{norm} = x_{0.25} - x_{0.75} \tag{1}$$

It is assumed that a low $IQR_{norm}$ indicates a high importance of the respective predictor variable, because it shows that many data points are concentrated in a short range of values. The whiskers also display certain percentiles. In this study they were modified to contain the data between the 25th and 12.5th percentile (lower whisker) and between the 75th and 87.5th percentile (upper whisker). Thus, box and whiskers contain the average 75% of the data points. This approach aims at using the average 50% and 75% of the wine-pressing facilities for the modelling. The remaining data points are treated as extreme values and outliers. Finally, this module ranks the variables by measures of their statistical dispersion and selects them by using threshold levels.

iii. Reclassification of predictor variable rasters:

In a next step, the percentile values are used to reclassify the predictor variable rasters. To every raster cell with a value inside of the box ($IQR_{norm}$) the new value of 2 was assigned, every cell with a value inside the whiskers received the new value of 1. Every cell outside the box and whiskers was set to the new value of 0. The described reclassification is applied to each topographical variable [51]. The reclassification for the cost distance variables was modified. Here, it can be assumed that the lowest costs (i.e., the lowest energy expenditures) are best for the location of a wine press since the facilities were preferably built close to settlements, roads etc. Therefore, all data points up to the 50th percentile (Median) were reclassified to 2, those within the 75th percentile to 1 and those above, i.e., points with high cost distances, to 0. It is assumed that the values within the box correspond to the actual well-suited value range for viticulture of the corresponding predictor variable. For the outliers, or anything outside box and whiskers, it is expected that these are actually marking the less favorable locations to settle a wine-pressing facility, which is why there are fewer data points to be found in that value range. However, the area where a site is located does not have to be completely unsuitable. Other location factors may be suitable in these places, which is why the facilities have settled there. In order to determine the overall suitability of each position in the research area, all predictor variables have to be considered in combination.

iv. Calculation of weightings:

This module computes a weighting factor based on the statistical importance of the respective predictor variable. One weighting used is a floating point factor after Ejstrud [52] which is based on the assumption that the datasets that have small standard deviations (*StdDev*) but an actually high dataset range, are the most important influential factors for the distribution of archaeological sites. It is calculated as follows for each predictor variable set:

$$w_E = \sqrt{\frac{1}{StdDev/Range}} \tag{2}$$

For datasets that contain a lot of outliers and that are skewed, a second, more robust, weighting factor was generated that is based on the $IQR_{norm}$:

$$w_{IQR} = 1 - IQR_{norm} \tag{3}$$

The greater the weighting factor, the greater the expected importance for the model. Each predictor variable raster is therefore multiplied by its weighting. The Python script also offers not to apply any weighting.

v. Generation of the suitability map:

In this module, the individual raster layers are combined to determine the overall suitability (S) of the study area. In principle, the different rasters are stacked and added up. The resulting raster is then normalized following Equation (1). The S raster shows floating point values between 0 and 1. High suitability (S) values correspond to areas where the location factors are favorable for the settlement of wine-pressing facilities.

vi. Model validation:

In order to evaluate the results, the model was validated and its performance and predicted power was estimated. Hence, it is checked how well the distribution of sites is described and tested if the model is overfitted, i.e., whether it is too closely aligned with the training data and therefore not able to perform well on a test or validation dataset. This module applies two different validation methods, an internal and an external. The internal validation tests how well the training data are described by the model, i.e., what percentage of sites gained an S value of ≥0.5. According to this, a high percentage indicates a good model performance. In addition, it has to be investigated how much area percentage is predicted as suitable. This is necessary because the model could simply predict the entire area as suitable and all sites would be included in this area. However, in this case the model is underfitted, as it does not make very precise predictions. As Verhagen [53] stated, a predictive model is only useful if it combines a high accuracy (equivalent to correct predictions) with a high precision (the ability to limit the area of high probability). Kvamme's gain statistics combines these two criteria. It determines the gain for an area of certain suitability from the area percentage of the modeled zone of interest ($p_a$) and the percentage of sites found within that zone ($p_s$) [3]:

$$Gain = 1 - \left(\frac{p_a}{p_s}\right) \tag{4}$$

The external model validation is carried out, based on a k-fold cross-validation [54] using a k of 5. If the percentage of site of S ≥ 0.5 has dropped considerably after cross-validation, it indicates some degree of overfitting.

The modules described above are executed consecutively. The flow chart of the algorithm, the expected input and its output is shown in Figure 3. The Python scripts and a detailed, structured code documentation of all modules can be accessed via https://github.com/lstubert/PreMo.

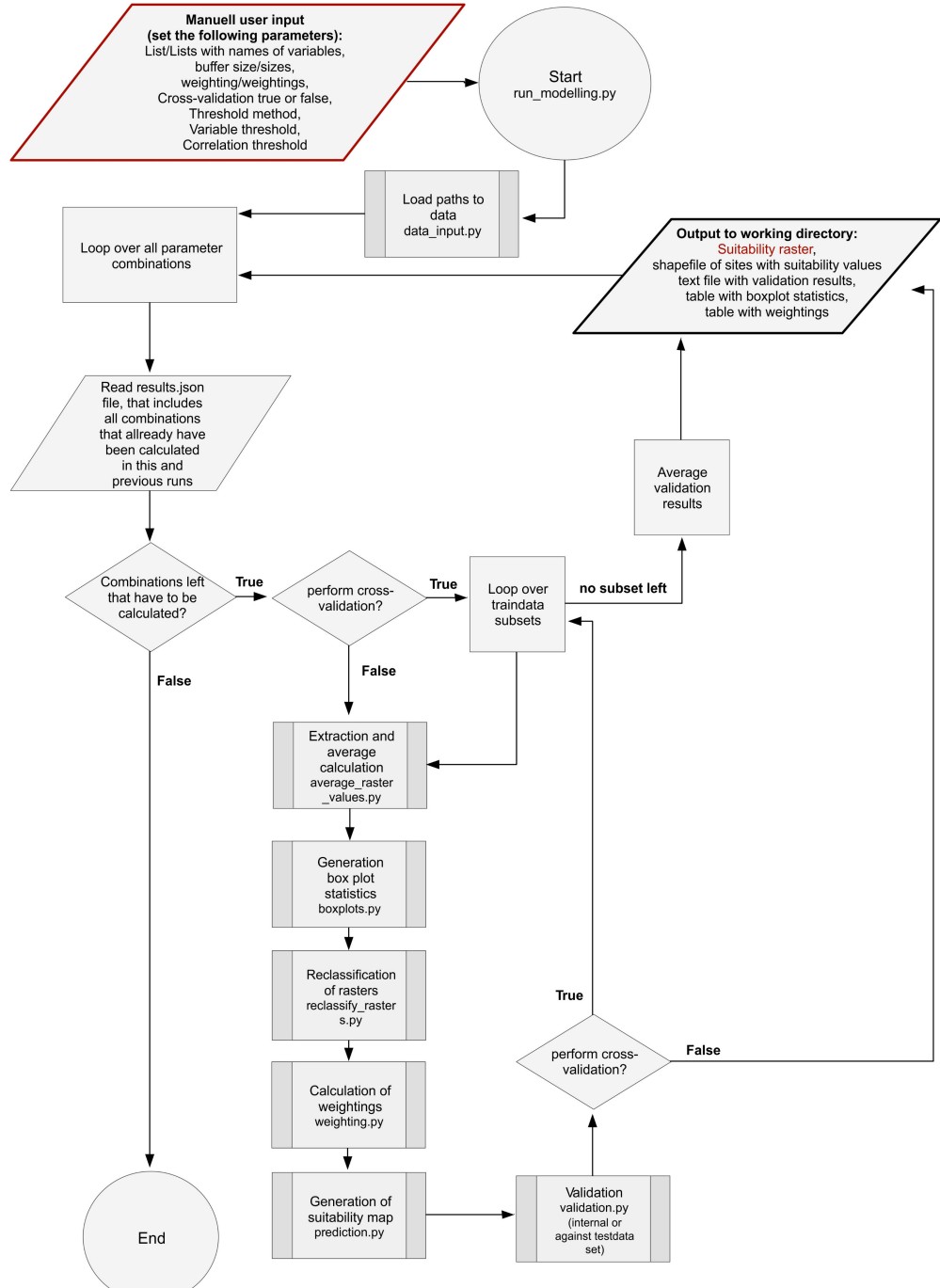

**Figure 3.** Flow chart of the main Python script. This script performs the predictive modelling by executing all modules.

### 2.4.2. Variable Selection by Expert Knowledge

In a first modelling approach, two models were calculated, that are based on *a priori* expert knowledge. The idea was to find a model for the spatial distribution of archaeological sites by using variables that performed well in other, similar studies.

The first model is based on the investigation of the ancient rural settlement structure in the hinterland of Pompeii (Campania, Italy) [8]. The following variables were used in this study: elevation, slope, aspect, curvature, vertical distance to channel network, LS-factor, topographic wetness index (TWI), topographic position index (TPI)-based landform classification and four cost distance variables

to different settlements and the river. The same variables were used here, with one modification concerning the TPI-based landform classification. This variable was substituted by the TPI itself, as landform classification was not cardinally scaled.

The second model is based on the predictor variables of a study about Roman viticulture in the lower Guadalquivir River area (*Hasta Regia* and *Gades territoria*) at *Hispania Citerior-Baetica* Roman province (Southwest Spain) [10]. The variables used are: wind, insolation, slope, soil types and distances to settlements, roads, sea, rivers and water supply. The same variable combination was used, with the exception of soil types as there was no soil map for the entire Laetanian Region available. However, since we assume a homogeneous geology in the study area, we hypothesize that soil types are represented by the topographic indices that are used as predictor variables. The variable water supply is, as Trapero Fernández [10] describes, difficult to establish, because there are many sources for water such as river water, groundwater or water from cisterns. In the present study we focused on the distance to rivers (cost distance to channel network).

### 2.4.3. Automated Variable Selection by Statistical Dispersion

In a second approach, the predictor variables are selected automatically, i.e., without *a priori* knowledge. At first, all predictor variables were analyzed by their descriptive statistics. The automatic selection is made on the basis of measures of statistical dispersion of the datasets, i.e., $IQR_{norm}$ and $w_E$. As already mentioned, it is assumed that a low $IQR_{norm}$ or a high $w_E$, respectively, indicates a high importance of the respective predictor variable. To enable automatic selection, the algorithm expects three additional entries from the user: the type of measure of statistical dispersion ($IQR_{norm}$ or $w_E$) that should be used and the specification of two threshold values. The first threshold indicates how many variables should be included in the model. The script ranks the variables after the chosen type of measure and then selects the appropriate number of the highest ranked variables. The second threshold determines the allowed Pearson's correlation coefficient (PCC) of inter-variable correlation to account for strongly correlated predictor variables. PCC is a statistic value that measures linear correlation between two variables X and Y. It has a value between +1 and −1, where 1 is total positive linear correlation, 0 is no linear correlation, and −1 is total negative linear correlation.

The script computes a correlation matrix and checks the variables for correlations. If the correlation threshold is exceeded, the variable with the lower ranking is sorted out. The selection of variables is then refilled by the next variable in the ranking. This iteration continues until the selected number of variables (specified by the threshold) is reached and none of them exceeds the correlation threshold.

### 2.5. Visualization of Results in an Interactive Web Map

In order to present the modelling results and make them available to the specialists, they should be visualized attractively and with as much additional value as possible. An interactive web map can be a good way to show connections and draw further conclusions. A draft of such a web map was created for the results obtained in the present study.

## 3. Results

### 3.1. Spatial Distribution of Archaeological Sites

Before performing the modelling, the distribution of the sites was visually evaluated (Figure 1). This first analysis suggests a spatial correlation. The sites are not dispersed in a regular pattern over the study area, nor do they seem to be randomly distributed. Instead, clusters can be identified, e.g., along the coast and around the cities of *Iluro* and *Barcino*. There are also more wine-pressing facilities near the roads and rivers, especially on lower-Llobregat-River area and its branch coming from *Egara*. It can be assumed that this distribution of sites is mostly influenced by topographical and socio-economic factors.

### 3.2. Predictive Modelling

In a first step of predictive modelling, a large number of terrain and socio-economic location characteristics had been prepared that potentially contributed to the distribution of archaeological sites. To avoid overfitting and to keep a model as simple as possible two different approaches of variable preselection were tested based on: (i) expert knowledge from similar research studies and (ii) measures of statistical dispersion.

### 3.2.1. Variable Selection by Expert Knowledge

One preselection was based on the work of Vogel et al. [8] a model of villae rusticae in the hinterland of Pompeii. Table 2 shows the predictor variable combination. The response variable was buffered with 100 m diameter and the weighting $w_E$ was used. Figure 9A illustrates the resulting suitability (S) map of model M1 showing only favorable areas with S ≥ 0.5. The result is reminiscent of the topographical scheme: the coastal region is predominantly classified as suitable, as is the area around the Llobregat River and the branched Vallesian Plain. An area with a comparatively large contiguous zone of high suitability (S ≥ 0.7), is the *centuriatio* plain or *ager barcinonensis*, located around ancient *Barcino's* colony [25]. At first sight, these correlations make a pretty positive impression on the model performance. However, a closer look reveals that the archaeological sites are rather situated on the edge of the high suitability areas. Moreover, in the northern coastal area, many sites could not be predicted by the model.

In the expert knowledge model M1 based on the study of Pompeii's hinterland (after Vogel et al. [8]) only 46% of sites were predicted with a suitability value of S ≥ 0.5 for the Laetanian Region. After cross-validation, this value decreases to 41%. The Pompeii model itself reached quite higher values of 76% (71% after cross-validation) in their prediction of rural settlements around Pompeii. The gain statistics in Figure 10 are also lower than in the Pompeii study. Therefore, it can be concluded that the corresponding variable combination that worked well to predict the distribution of *villae rusticae* around Pompeii holds not as much explanatory value to model the distribution of winegrowing facilities in the Laetanian Region. Therefore, a better adapted model has to be found for the specific topic of viticulture in this research area.

Expert knowledge model M2 based on the study of the lower Guadalquivir territory in Southwest Spain and focusing exclusively on Roman viticulture (after Trapero Fernández [10]), seems obtain better suitability results with its variable combination. To keep the models comparable we also used 100 m buffered sites and $w_E$ as weighting. The results of M2 applied to the Laetanian Region are shown in Figure 9B. It is striking that spatial patterns of high suitability zones resemble the results of M1. However, compared to the Pompeii model, the area ratio that is shown with high S values is larger. The high suitability around *ager barcinonensis*, which were already noticed in the Pompeii model, became even more pronounced. However, also significant differences can be seen in the northeastern Laetanian coastal area. This time, this area is partially classified as suitable. Also, some zones in the Vallesian Plain received a high S values. Thus, especially in the north of the research study region, many sites were predicted correctly. In total, for 77% of sites the model predicted a value of S ≥ 0.5. After cross-validation this reduced to 74%. This shows a significantly better performance of M2 (based on the Southwest Spain model) focused in the specific topic of Roman viticulture than M1 (based on the Pompeii model) focused on Roman agrarian production in general.

The results so far show that the preselection of variables should be adapted to the respective area and intended use. This usually requires reliable expert knowledge about the region and the research subject. Trapero Fernández [10] for example refers in his study to *Lucius Junius Moderatus Columella*, the prominent Roman agronomist and writer and his texts and knowledge about important environmental factors that should be considered for the establishment of an agricultural farm.

**Table 2.** Variable importance ranking of the predictive models.

| Rank | Predictor Variable | Statistical Importance [%] | |
|------|--------------------|---------------------------|---|
| **M1 - Expert: Pompeii Study** | | | |
| 1 | Curvature | 100 | |
| 2 | TPI | 99.3 | |
| 3 | LS-factor | 92.1 | |
| 4 | Elevation | 91.8 | |
| 5 | Vertical distance to channel network | 89.4 | |
| 6 | Cost distance to rivers | 87.4 | |
| 7 | TWI | 86.7 | |
| 8 | Cost distance to settlements | 86.5 | |
| 9 | Aspect | 83.7 | |
| 10 | Slope | 83.2 | |
| **M2 - Expert: Southwest Spain Study** | | | |
| 1 | Direct insolation | 100 | |
| 2 | Cost distance to roads | 90.4 | |
| 3 | Cost distance to rivers | 82.7 | |
| 4 | Cost distance to secondary settlements | 81.7 | |
| 5 | Wind | 80.6 | |
| 6 | Cost distance to primary settlements | 80.4 | |
| 7 | Slope | 78.6 | |
| 8 | Cost distance to coast | 68.9 | |
| **M3 - Automated: $IQR_{norm}$** | | | |
| 1 | Profile curvature | 100 | |
| 2 | Direct insolation | 96.0 | |
| 3 | Cost distance to favorable hillsides | 91.0 | |
| 4 | Cost distance to roads | 88.0 | |
| 5 | Cost distance to secondary settlements | 84.9 | |
| 6 | Vertical distance to channel network | 84.8 | |
| 7 | Cost distance to rivers | 82.7 | |
| 8 | LS-factor | 82.4 | |
| 9 | Cost distance to primary settlements | 77.3 | |
| 10 | Cost distance to coast | 49.1 | |
| **M4 - Automated: $w_E$** | | | |
| 1 | Direct insolation | 100 | |
| 2 | Curvature | 94.4 | |
| 3 | Cost distance to roads | 90.4 | |
| 4 | Diffuse insolation | 88.8 | |
| 5 | LS-factor | 86.9 | |
| 6 | Vertical distance to channel network | 84.4 | |
| 7 | Cost distance to rivers | 82.7 | |
| 8 | Cost distance to secondary settlements | 81.6 | |
| 9 | Cost distance to primary settlements | 80.3 | |
| 10 | Cost distance to coast | 68.9 | |

### 3.2.2. Automated Variable Selection by Statistical Dispersion

In the following, the selection of predictor variables will be based on measures of their statistical dispersion. By stepping away from the need of expert knowledge and developing an objective approach for variable selection it is possible to simplify and automate the predictive modelling process.

The results of the reclassified rasters of topographic predictor variables are shown in Figure 4. It is noticeable that most of the topographical parameters show striking similarities. The central coast appears more or less as a white band in the vast majority of topographic rasters, because it is largely reclassified as suitable (value 2). The mountainous region behind the coast was mostly classified as unsuitable (value 0) and, therefore, appears as a black area. The region behind, the Vallesian Plain, shows branch-shaped white patterns of areas classified as suitable, which is connected to the coast by a passage located in the north of Barcelona's Plain. How strongly these suitable areas are connected differs from raster to raster. The elevation raster, for example, shows strongly contiguous areas, since the height generally increases rather gradually in the terrain. Other variables, on the contrary, which have strong fluctuations in small distances, show much more noisy reclassifications, for example curvature and TPI. A closer look reveals another difference that only applies to the Vallesian Plain. Although this plain is often reclassified as suitable, some rasters in this area seem to be some kind of negatives of other rasters. An example is the LS-factor and the direct insolation. The LS-factor raster has large unsuitable areas correlated to the positions of rivers in the plain. In the case of direct

insolation, on the other hand, exactly these areas are shown to be suitable. These features and patterns are less obvious in the distance to the channel network raster. In the aspect raster, spatial patterns can only be guessed. The southeastern slopes of the Central Coast region, which are reclassified as suitable, are slightly noticeable. In general, it can be summarized that some of the variable rasters could be sorted into groups of similar appearance. The results of visual analysis are supported by the correlation matrix (Figure 6). It shows that some variables are strongly correlated. This applies to slope, LS-factor and TRI but also curvature, profile curvature and TPI received high Pearson's correlation coefficients (PCC). Thus, it is reasonable to exclude the correlated variables from the modelling.

Figure 5 shows the reclassified rasters of cost distances to the different socio-economically relevant locations. Comparing the coast and the roads rasters, it is noticeable that one of the splitting arms of the Via Augusta, the main ancient road of the region, runs close to the coastline. In that respect, these two variables show largely overlapping spatial features. Around the settlements, larger areas of suitability arose that show a very irregular shape. Here, it can be seen again that the topography in the region is strongly textured and not easily accessible.

There is no particularly strong correlation between the cost distances (Figure 6), thus, they can all be used for modelling. A high correlation of 0.86 exists between the elevation and distance to the coast. Therefore, one of these variables should be removed as predictor variable.

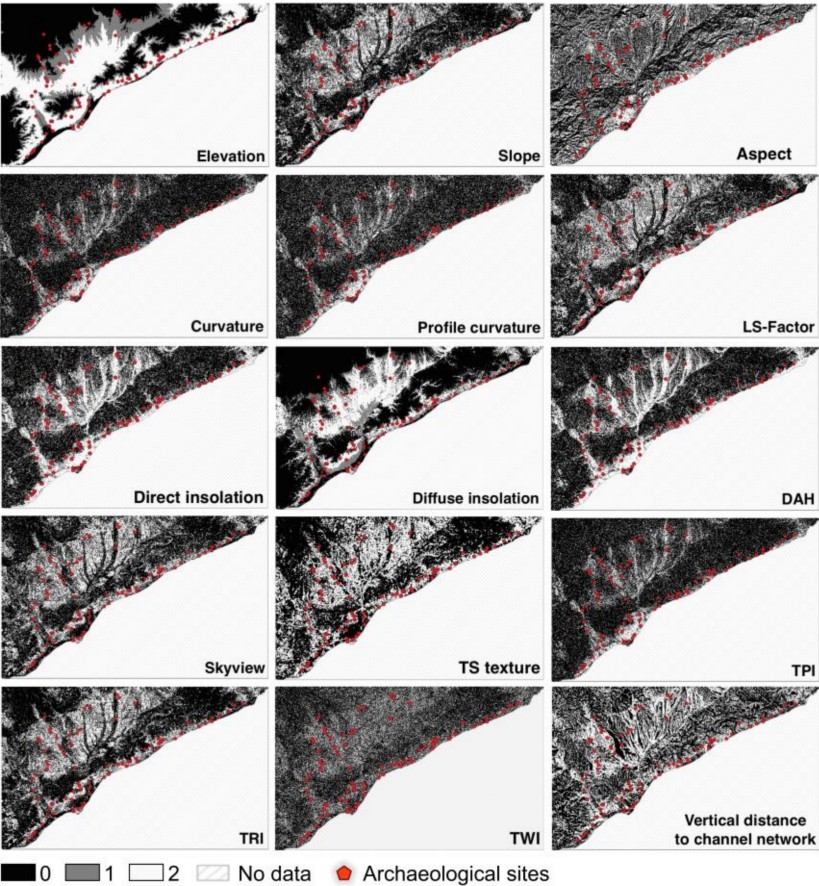

**Figure 4.** Rasters of the reclassified topographic predictor variables.

As described earlier, it is assumed that the variables that control the distribution of sites tend to have a lower $IQR_{norm}$ than variables which are of no effect. As can be seen in Figures 7 and 8, the $IQR_{norm}$ varies significantly for the different topographical variables. Curvature, profiles curvature, direct insolation, DAH and TPI show comparatively lower $IQR_{norm}$. For these variables, most sites occur in a lower value range. The lowest $IQR_{norm}$ of all variables was reached by the profile curvature with 0.09.

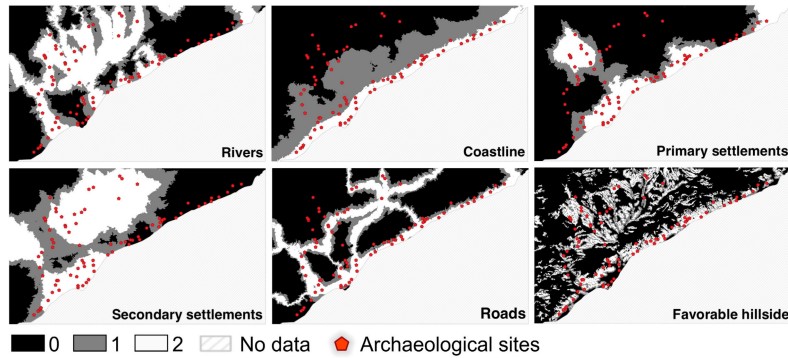

**Figure 5.** Rasters of the reclassified cost distance predictor variables.

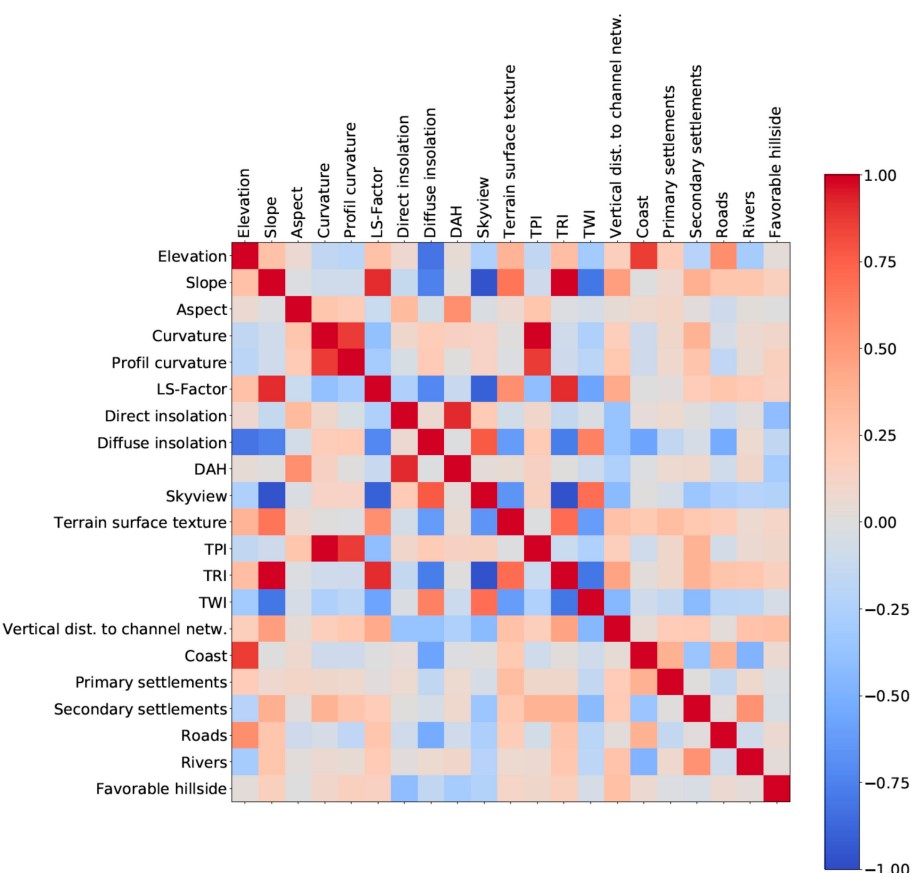

**Figure 6.** Correlation matrix of predictor variables showing Pearson's correlation coefficients.

As can be seen in Figure 8, the medians of distance variables tend to be lower than the medians of topographic variables. This is due to the fact that proximity or not too great separation to all tested socio-economically relevant locations is preferred over great distances. For the cost distances, the variability of $IQR_{norm}$ is much lower compared to the topographic variables. This can be an indication that the various socio-economic variables are of similar importance for the location of the wine pressing facilities. The only exception is the distance to the coast. At 0.55, it has the largest $IQR_{norm}$ of all variables. On that basis it could be argued to omit the coast from the modelling. A low median for the roads and favorable hillsides shows that the locations in immediate proximity were most popular. The median for the coast and rivers, however, is slightly higher indicating that wine-pressing facilities were preferably built in some distance, probably not to be affected by floodings. This phenomenon was also observed for *villae rusticae* around Pompeii [8]. For settlements, the median

distance was even higher since the *villae* with wine presses and vineyards were rather located in the surrounding countryside.

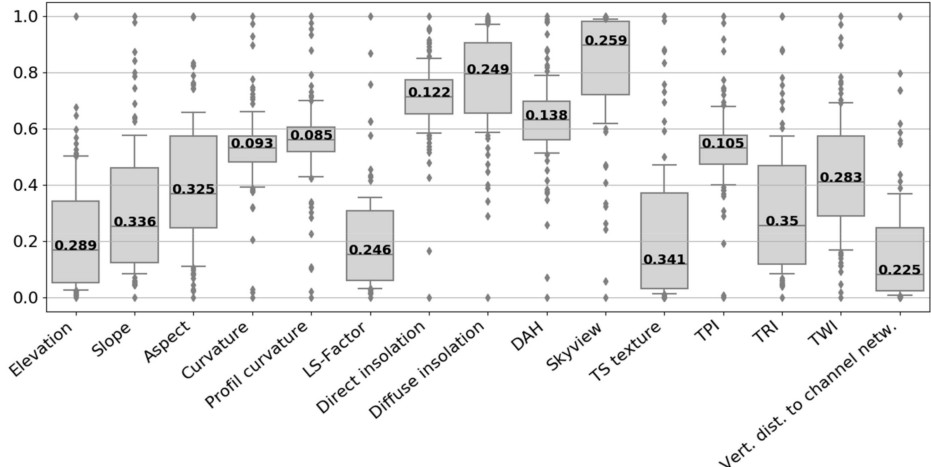

**Figure 7.** Boxplots of topographic location characteristics of sites on a normalized scale. The whisker length is modified to that effect, that they correspond to the data between the 12.5th and 87.5th percentiles. The value written in the box shows the interquartile range (*IQR*$_{norm}$) of the respective dataset.

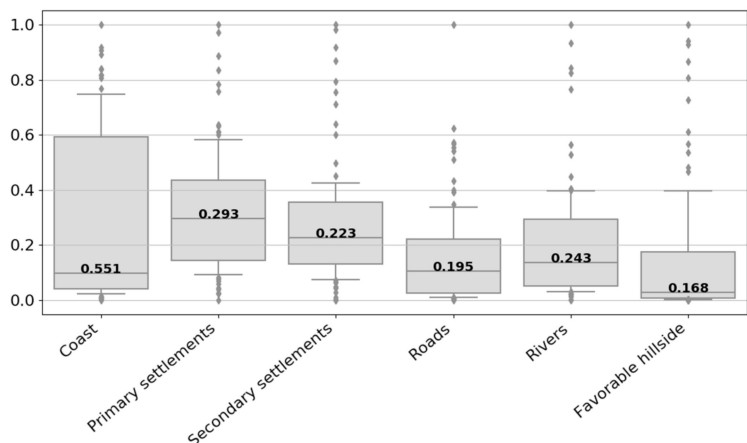

**Figure 8.** Boxplots of cost distance location characteristics of sites on a normalized scale. The whisker length is modified to that effect, that they correspond to the data between the 12.5th and 87.5th percentiles. The value written in the box shows the *IQR*$_{norm}$ of the dataset.

An alternative method for variable importance ranking and weighting is the calculation after Ejstrud [52] ($w_E$, Equation (2)). Table 3 illustrates the descriptive statistics of the predictor variables as well as their importance ranking by $w_E$. The highest $w_E$ was obtained by direct insolation, the lowest by the cost distance to the coast. The variables direct insolation, curvature, TPI, DAH and profile curvature that ranked highest in the *IQR*$_{norm}$ ranking are also represented by high predictive values. This confirms their importance to predict ancient wine-pressing facilities in the study region. It is noticeable, that curvature and TPI also received the highest importance in the prediction of *villae rusticae* around Pompeii [8]. The socio-economic parameter with the highest ranking was the cost distance to roads.

Generally, it can be said that the *IQR*$_{norm}$ ranking differs slightly from the $w_E$ ranking. In one case the difference is particularly strong. The cost distance to favorable hillsides has a very low *IQR*$_{norm}$, so it should be advantageous for the model. At the same time it received the third worst $w_{E,}$ value. This can be explained by a more in-depth look at the data. Table 3 shows that the minimum value of distance to favorable hillsides is 0, the mean value is only 22 and the maximum value is as high as 763.

A lot of wine pressing facilities have to be located in an area that is either exactly in or very close to a favorable hillside. This is also confirmed by the reclassified raster in Figure 8. Hence, the data are very different from a normal distribution. Instead, they show a left skewness, which means that the specification of a standard deviation and, therefore, the resulting $w_E$ can be misleading. This effect must also be kept in mind when considering the other variables, which also deviate to some extent from the normal distribution. This applies to all cost distances, but also some topographic variables, for example skyview. Consequently, the $IQR_{norm}$ that is robust towards skewed data seems to be the more appropriate measure of variable importance as it is not dependent on normal distribution of the data.

**Table 3.** Descriptive statistics of the predictor variables. The table is sorted by decreasing $w_E$.

| Predictor Variable | Min. | Mean | Max. | Std.Dev. | $w_E$ |
|---|---|---|---|---|---|
| Direct insolation | 1.81 | 3.26 | 3.85 | 0.31 | 2.58 |
| Curvature | −0.02 | 0.00 | 0.02 | 0.00 | 2.44 |
| TPI | −0.33 | −0.01 | 0.28 | 0.10 | 2.42 |
| DAH | −0.28 | 0.01 | 0.18 | 0.08 | 2.41 |
| Profile curvature | −0.01 | 0.00 | 0.00 | 0.00 | 2.37 |
| Cost distance to roads | 14.00 | 434.00 | 3962.00 | 724.63 | 2.33 |
| Diffuse insolation | 0.83 | 0.88 | 0.89 | 0.01 | 2.29 |
| LS-factor | 0.09 | 2.74 | 17.50 | 3.46 | 2.24 |
| Elevation | 0.22 | 77.31 | 460.72 | 91.78 | 2.24 |
| Vertical distance to channel network | 0.07 | 4.46 | 53.79 | 11.31 | 2.18 |
| Cost distance to rivers | 22.00 | 887.00 | 6261.00 | 1364.46 | 2.14 |
| TWI | 3.00 | 7.30 | 11.10 | 0.97 | 2.12 |
| Cost distance to secondary settlements | 9.00 | 4070.00 | 17,881.00 | 4017.31 | 2.11 |
| Skyview | 0.97 | 1.00 | 1.00 | 0.01 | 2.11 |
| Cost distance to primary settlements | 42.00 | 2998.00 | 9995.00 | 2311.47 | 2.08 |
| Aspect | 59.07 | 146.09 | 295.74 | 56.72 | 2.04 |
| TRI | 0.06 | 0.78 | 2.88 | 0.68 | 2.03 |
| Slope | 0.42 | 5.23 | 19.54 | 4.65 | 2.03 |
| Cost distance to favorable hillsides | 0.00 | 22.00 | 763.00 | 191.70 | 2.00 |
| TS texture | 0.00 | 1.31 | 11.02 | 2.86 | 1.96 |
| Cost distance to coast | 77.00 | 1043.00 | 10,012.00 | 3137.40 | 1.78 |

Finally, several models were calculated based on different settings including:

i.     The two described methods for variable importance ranking.
ii.    Different thresholds for inter-variable correlations.
iii.   Different numbers of predictor variables.
iv.    Different buffer area sizes of 50, 100 and 150 m.
v.     Variable importance-based weighting or no variable weighting.

The detailed results of these different model iterations are shown in the Appendix A (Tables A1 and A2). The comparison of their gain values has shown that for both variable importance methods the correlation threshold that received the best model performance was 0.75. Furthermore, a buffer area of 100 m and a variable importance-based weighting produced the best results. The influence of the number of variables on the model performance varied to certain extend. However, it can be said that models with five topographic variables have shown good results. This is a logically justifiable amount of variables to keep the models as simple as possible.

As illustrated in Figure 9C,D, the $IQR_{norm}$ model predicted 68% of sites with S ≥ 0.5, compared to 67% for the $w_E$ model. After cross-validation, however, this percentage was still 68% for the $IQR_{norm}$ model and decreased to 60% for the $w_E$ model. This points to some degree of overfitting of the $w_E$ model whereas the $IQR_{norm}$ model is rather robust. The gain statistics of both models is not strikingly different (Figure 10). Both models were not able to predict any site within the highest class of S ≥ 0.9.

Similar to the Southwest Spain model (M2), the automated models predicted vast areas of the coastal region as highly suitable. Also, some areas in the Vallesian Plain received rather high S values. In general, the automated models predict fewer areas as very highly suitable with S ≥ 0.8. Finally, it can be said that the $IQR_{norm}$ and $w_E$ model look rather similar showing only small-scape disparities.

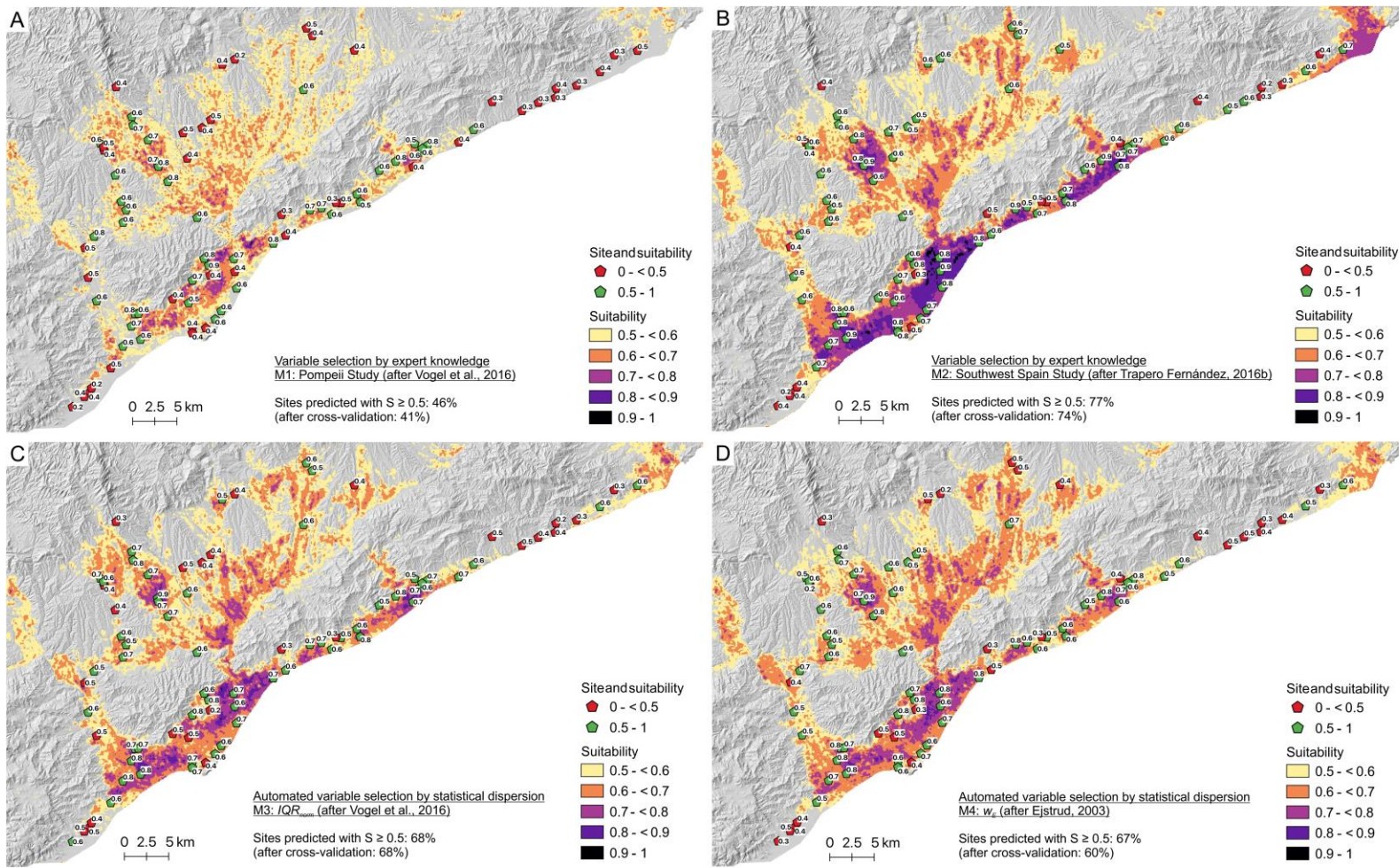

**Figure 9.** Suitability maps of predictive models with variable selection based on expert knowledge (**A**: M1—Pompeii Study (after Vogel et al. [8]); **B**: M2—Southwest Spain Study (after Trapero Fernández [10])) and on statistical dispersion (C: M3—*IQR_norm* (after Vogel et al. [8]); D: M4—*w_E* (after Ejstrud [52])). Labels on white background refer to rounded predicted S values for the respective archaeological sites.

### 3.3. Interactive Web Map

Because of its best performance, the Southwest Spain model (Figure 9B) was chosen and visualized for further analysis using an interactive web map. It can accessed via https://github.com/lstubert/lstubert.github.io. Within the map, individual layers can be displayed or deselected via the legend. By clicking on an archaeological site in the map, pop-up windows with additional information to the respective objects open. The suitability raster is separately addressable in each value range. Furthermore, several filters were integrated: e.g., the sites can be displayed according to the duration of their usage, their estimated wine presses, and their initial date.

## 4. Discussion

The gain statistics of the four predictive models are illustrated in Figure 10. Comparing the spatial representation of suitability (S) values in the different models, it is striking that, in general, the main spatial patterns recur. However, the S values reached in M1 are much lower compared to the other three models. In general, M1 shows the lowest model performance. Only 46% of archaeological sites could be predicted in suitable areas. This low performance can be explained by comparing the topographies of the two study regions. The hinterland of Pompeii is largely a plain. In contrast, the Laetanian Region is characterized by great differences in altitude and topography. The flat coastal area is located directly south of a mountain range with heights of up to 750 m a.s.l. In the North follows the Vallesian Plain. The three main rivers, some affluents and a huge number of local streams dissect the pre-littoral and littoral mountain ranges and form longitudinal belts of alluvial sedimentation on the seashores. This varying topography makes movement much more difficult compared to a plain. However, movement was most likely a decisive factor in the choice of locations for vineyards and wine-pressing facilities. This accessibility to a certain location can be quantified by the calculated cost distances. M1, however, only included cost distances to two different socio-economic locations: settlements and rivers.

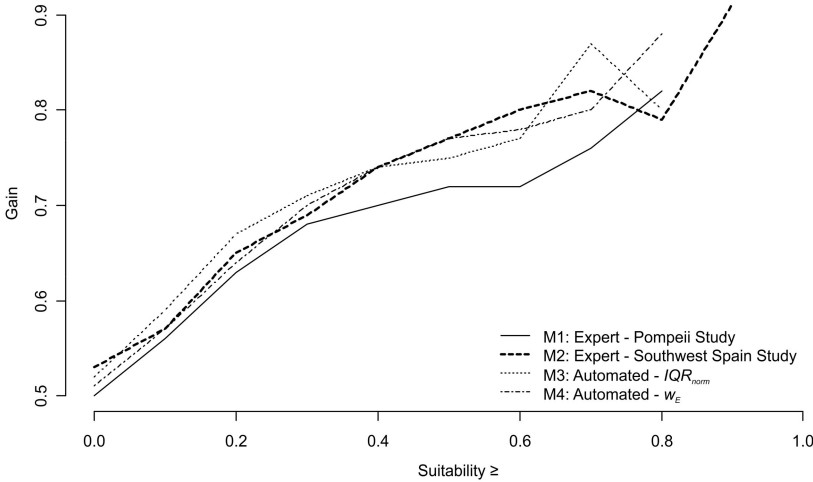

**Figure 10.** Gain versus suitability plot of the predictive models. The tabulated gain statistic can be found in the Appendix A (Table A3).

An indication that the consideration of more cost distances is increasing the model performance, is given by comparison with model M2, where the distance to roads and coastline were additionally included. The variable combination of M2 was taken from a specific study on Roman viticulture in Southwest Spain [10]. Thus, spatially and thematically it is very close to the subject of this paper. M2 showed much higher gain values. A total of 77 % of sites were predicted in suitable area, which can be rated as a good result, considering that the modelling approach only aimed at modelling the average 75% of archaeological sites. The fact that this percentage did not decrease considerably after cross-validation (74 %) indicates that the model is only slightly over-fitted and relatively robust.

The variable combination derived from this model can therefore describe the distribution of the viticulture facilities in the Laetanian Region considerably well. The great difference in explanatory value of M1 and M2 leads to the conclusion that the predictor variables should be carefully selected for the respective topic and region of interest. Hence, models that have worked well in a determinate geospatial context of agrarian activity cannot be easily applied to other datasets and study areas.

Whereas M1 and M2 depended on expert knowledge, the variable selection of the models M3 and M4 was automated based on measures of statistical dispersion. At first sight, M2 as well as M3 and M4 show great similarities in the spatial distribution of suitable areas, especially in the Vallesian Plain. All three models have the branched shape of suitable areas and also the particularly high S values are located in the same area. The mountain region between the coast and the plain hardly shows any areas with S ≥ 0.5. One exception is for example the area north of *Iluro*. At the coast, M2 classified a more contiguous area as highly suitable (S ≥ 0.7) whereas the two automated models show rather differentiated S values. In general, M3 and M4 predicted fewer areas to be highly suitable. In contrast, M2 received higher gain values for S ≥ 0.9, but lower gains for S ≥ 0.8. Cross-validation has shown that M1, M2 and M4 display some degree of overfitting which is highest for M3. In contrast, no overfitting was observed for M3.

The choice of the best-performing model can be finally based on different measures. However, if the objective is to predict sites correctly in particular in the highest suitability classes or to predict the highest quantity of sites correctly in suitable locations (S ≥ 0.5), M2 is the most accurate model. This result indicates that winegrowing activity has specific demands on the location, which the automated models were not completely able to cover. The fact that M2 received the best model performance is particularly noticeable because it refers to *Lucius Junius Moderatus Columella*, the prominent Roman agronomist and writer and his texts and knowledge about important environmental factors that should be considered for the establishment of an agricultural farm [10,55].

Concerning the automated models, selecting variables by $IQR_{norm}$ performed slightly better than selecting by $w_E$ after Ejstrud [52]. This might be due to skewed and not normally distributed datasets, especially among the cost distance data. In principle, a test for normal distribution should be carried out for each variable before calculating the standard deviation. Otherwise it is recommended to use quantile ranges [56].

To conclude, it can be noted that automatic variable selection by statistical dispersion metrics, especially by the more trustworthy $IQR_{norm}$, was successful. The automated models served well as a first approximation of suitability modelling when *a priori* knowledge on operating processes and interactions influencing the spatial distribution of sites is not available. Consequently, this modeling approach can be used in particular for theory building.

It can be deduced from the models that the topography has a very large influence on the distribution of archaeological sites in the study region. However, it is not only the terrain characteristics themselves that are decisive, but also their effects on movement and the distribution of goods in the terrain, which are expressed by the cost distance variables. The models show that in general the most important exclusion criterion for wine-pressing facilities is the avoidance of the mountain regions and steep slopes. On the contrary, high suitability zones can be found in valleys and in the lowlands along the coast. The prediction results in particular correspond to the expectations set up before the modelling and can at least partly be explained hydrologically and geomorphologically. Steep slopes of more than 15 % are not only a natural constrain due to their difficult accessibility. It is also conceivable that planting vines in steep slopes were not recommended, because denudation processes prevent the deposition of an agriculturally usable soil cover, unless you can build cultivation terraces which would involve great investment of resources and maintenance effort [47]. Highly elevated regions are also not suitable, even if they have less steep slopes as the groundwater table is no longer accessible which could be the reason why the models did not predict any suitable areas >454 m a.s.l. (referring to the ancient DEM).

The higher suitability in the coastal area and in particular around the coastal towns coincides quite well with a higher occurrence of sites. This confirms the assumption that the cities were important nodes for the distribution of wine and its derivates, especially for the interregional and overseas market.

For M2, 19 out of 82 sites were not predicted correctly. All of these sites occurred around 25 BC. At that time, there was a huge upswing in the production of Laetanian wine [13]. The settlement pressure at that time could have been so high that these wine-pressing facilities had to settle in these marginal areas. However, it can also be explained by a winegrowing intensification process to expand the production area to the coastal foothills for covering a rising demand. Furthermore, a specialization process could have resulted in the production of different sorts and qualities of wine that have different demands on the growing location. Ancient Roman agronomists tell us that some vines and grape varieties were cultivated well in the plain, and others, which are of higher quality, have to grow on steeper slopes for receiving more sunlight for its proper maturation process [55,57].

Martín i Oliveras and Revilla [20] stated that one Catonian wine press can process between 360 and 528 *iugera* of vineyards in 30 days to 44 days which is the duration of the *vindemia* period indicated by Roman agronomists. The *iugerum* is an agrarian unit of surface commonly used in Roman times which corresponds to $\cong 2518$ m$^2$. Consequently, one Catonian wine press refers to 90 to 132 hectares of vineyards. With this knowledge, and the modeled high suitability areas, it was possible to calculate how many Catonian wine presses have approximately been needed for processing a maximum potential vineyard crop yield in this territory. The area calculated by model M2 with $S \geq 0.5$ is 64,925 hectares. Considering this whole area, between 492 to 721 Catonian wine presses were needed for processing these potential vineyards crop yields. If it is assumed, that most wine pressing facilities had two presses, there was a maximum potential capacity of between 246 and 360 of such winemaking facilities. The same calculation was done with $S \geq 0.75$ which amounts to a total area of 9054 hectares. This would result in 68 to 100 winemaking facilities with a single Catonian wine press and thus between 34 and 50 winemaking facilities with two Catonian wine presses. This second quantification is clearly an underestimation, because the minimum number of winemaking facilities that have actually been excavated and documented in this region is 82. From the highest estimated number of potential wine-pressing facilities, i.e., 360, the 82 known sites were subtracted. The remaining 278 potential sites, also equipped with a minimum of two Catonian wine presses, were then randomly distributed over the high suitability areas, considering a minimum distance of 500 meters between two sites which corresponds to the shortest known inter-site distance. The result is shown in Figure 11.

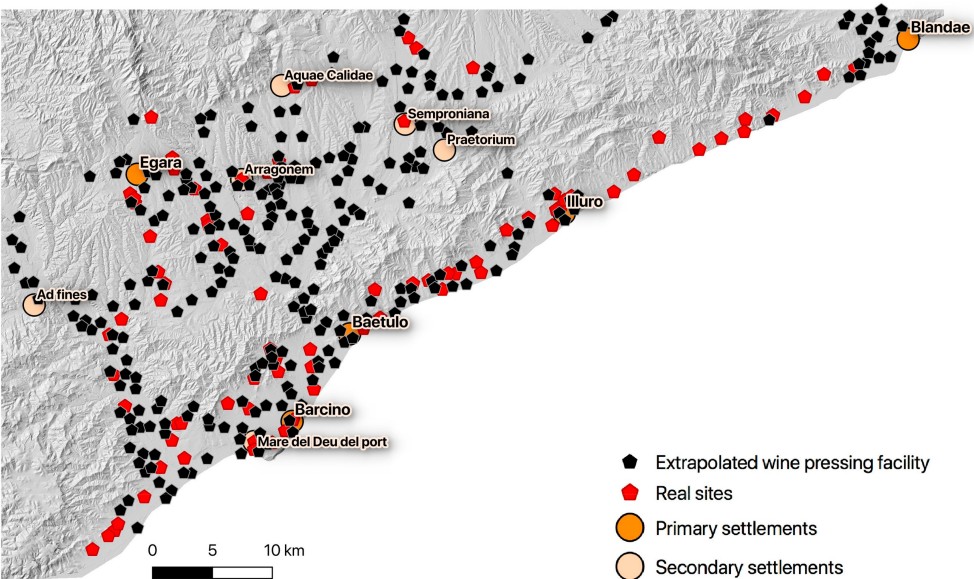

**Figure 11.** Distribution of 360 potential wine pressing facilities over the suitability area $S \geq 0.5$, predicted by M2.

However, the suitable areas were probably also used for other crops than vineyards, which could be one of the reasons why the actual number of wine presses may be smaller, than calculated. Moreover, in reality, the density of sites would most likely be dependent on the S value with higher site densities for higher suitabilities.

## 5. Conclusions

The general objective of the present paper was to calculate the suitabilities for archeological sites in the Laetanian Region related to intensive and specialized Roman viticulture in order to determine the underlying factors of their distribution. Therefore, the archaeological dataset of 82 documented wine-pressing facilities was used as response variable in a predictive modelling approach. Furthermore, 15 topographical and 6 socio-economic location characteristics were taken into account, which may have had, as stated by other authors, an influence on the distribution of the sites. In a first step, two models were developed where the variable selection was based on expert knowledge from previous studies on Roman agriculture and viticulture. However, this work additionally aimed at semi-automating the process of variable selection. Hence, the variables were selected using statistical distribution metrics. The resulting models with automated variable selection showed good performance. They serve well as a first approximation of suitability modelling when no a priori knowledge on operating processes and interactions influencing the spatial distribution of sites is available. Consequently, this modeling approach can be used for theory building. However, it is of special interest that the best prediction performance was obtained by an expert knowledge model utilizing a combination of predictor variables that is based on the specific recommendations on viticulture by *Lucius Junius Moderatus Columella*, the prominent ancient Roman agronomist. The model was used to make the first assumptions and theories about the underlying factors that had an impact on the development of viticulture in the Laetanian Region. The results indicate that the accessibility of a location and its connectivity to the local and regional distribution centres and trade routes, determined by terrain steepness, was decisive for the settlement of winemaking facilities. Moreover, on the basis of the predictive model and findings on experimental archaeology, the maximum number of winegrowing facilities that could have existed in the Laetanian region in Roman times was extrapolated to 360. As the applied modeling approach focusses on the average 50% and 75% of wine pressing facilities, some site locations could not be explained by the predictive model. Thus, future research should particularly focus on those locations to find reliable hypotheses for settlement under these "marginal" and "less suitable" conditions. This is expected to provide more detailed insights into the complexity of ancient rural settlement and viticulture in the Laetanian Region.

The results of the best model were visualized in a web map for interactive evaluation. In the next step, the predictive model should be comprehensively used by archaeologists to determine which additional hypotheses can be derived and how these can be integrated into the current state of research. An important question would be whether the results are compatible with previous assumptions and findings, or whether contradictions arise.

To simplify the technical aspects of modelling even more, a lightweight graphical user interface could be written where the user can set or select the model. This makes the application easier for people who have little or no experience with programming scripts, since the actual code no longer needs to be opened and changed. It would also be conceivable to integrate such an interface into QGIS in terms of a QGIS plugin.

Finally, it should be emphasized that the application of this predictive modelling algorithm is not limited to the specific field of archaeology. In contrast, this methodology may also be used in other scientific fields, such as ecology to model species distributions or in economic geography for production site analysis.

**Author Contributions:** Conceptualization, A.M.i.O., L.S. and S.V.; methodology, L.S. and S.V.; software, L.S.; validation, L.S.; formal analysis, L.S.; investigation, L.S.; data curation, A.M.i.O.; writing—original draft preparation, L.S. and S.V.; writing—review and editing, A.M.i.O., M.M., H.S.; visualization, L.S. and S.V.; supervision, A.M.i.O., M.M., H.S. and S.V. All authors have read and agreed to the published version of the manuscript.

**Funding:** This research received no external funding

**Conflicts of Interest:** The authors declare no conflict of interest.

## Appendix A

**Table A1.** Table of model results calculated to test and compare thresholds. (Due to geometry restrictions the header items are abbreviated and stated here: 1—Variable input, 2—Buffer size, 3—Weighting, 4—Variable Threshold, 5—Correlation Threshold, 6—% Good Prediction, 7—Gain ≥ 0.5, 8—Gain ≥ 0.75, 9—Threshold Type).

| 1 | 2 | 3 | 4 | 5 | 6 | 7 | 8 | 9 |
|---|---|---|---|---|---|---|---|---|
| all | 100 | - | 5 | 0.75 | 62.2 | 0.76018 | 0.89358 | $w_E$ |
| all | 100 | - | 6 | 0.75 | 60.98 | 0.76649 | 0.88189 | $w_E$ |
| all | 100 | - | 5 | 0.6 | 59.76 | 0.75384 | 0.83667 | $w_E$ |
| all | 100 | - | 4 | 0.75 | 64.63 | 0.77161 | 0.82079 | $w_E$ |
| all | 100 | - | 4 | 0.6 | 59.76 | 0.75805 | 0.78433 | $w_E$ |
| all | 100 | - | 5 | 0.9 | 57.32 | 0.77141 | 0.77412 | $w_E$ |
| all | 100 | - | 4 | 0.9 | 47.56 | 0.73512 | 0.75365 | $w_E$ |
| all | 100 | - | 6 | 0.9 | 59.76 | 0.76349 | 0.72244 | $w_E$ |
| all | 100 | - | 6 | 0.6 | 56.1 | 0.74898 | 0.67689 | $w_E$ |
| all | 100 | - | 4 | 0.75 | 63.41 | 0.78526 | 0.81762 | $IQR_{norm}$ |
| all | 100 | - | 4 | 0.6 | 63.41 | 0.78526 | 0.81762 | $IQR_{norm}$ |
| all | 100 | - | 4 | 0.9 | 53.66 | 0.78484 | 0.79106 | $IQR_{norm}$ |
| all | 100 | - | 5 | 0.75 | 65.85 | 0.7893 | 0.88384 | $IQR_{norm}$ |
| all | 100 | - | 5 | 0.6 | 65.85 | 0.7893 | 0.88384 | $IQR_{norm}$ |
| all | 100 | - | 5 | 0.9 | 53.66 | 0.77443 | 0.76765 | $IQR_{norm}$ |

**Table A2.** Table of model results calculated to test and compare buffer sizes, weightings and variables. (Due to geometry restrictions the header items are abbreviated and stated here: 1—Variable input, 2—Buffer size, 3—Weighting, 4—Variable Threshold, 5—Correlation Threshold, 6—% Good Prediction, 7—Gain ≥ 0.5, 8—Gain ≥ 0.75, 9—Threshold Type).

| 1 | 2 | 3 | 4 | 5 | 6 | 7 | 8 | 9 |
|---|---|---|---|---|---|---|---|---|
| all | 100 | - | 6 | 0.75 | 68.29 | 0.78024 | 0.8724 | $IQR_{norm}$ |
| all | 100 | - | 6 | 0.6 | 62.2 | 0.78863 | 0.87908 | $IQR_{norm}$ |
| all | 100 | - | 6 | 0.9 | 57.32 | 0.78219 | 0.81731 | $IQR_{norm}$ |
| all | 100 | - | 5 | 0.75 | 62.2 | 0.76018 | 0.89358 | $w_E$ |
| Without hill | 100 | - | 5 | 0.75 | 62.2 | 0.76018 | 0.89358 | $w_E$ |
| all | 100 | $w_E$ | 5 | 0.75 | 67.07 | 0.75477 | 0.87128 | $w_E$ |
| Without hill | 100 | $w_E$ | 5 | 0.75 | 67.07 | 0.75477 | 0.87128 | $w_E$ |
| Without coast | 100 | - | 5 | 0.75 | 60.98 | 0.7386 | 0.82593 | $w_E$ |
| Without coast | 100 | $w_E$ | 5 | 0.75 | 63.41 | 0.73364 | 0.82593 | $w_E$ |
| all | 250 | - | 5 | 0.75 | 48.78 | 0.78253 | 0.80205 | $w_E$ |
| Without hill | 250 | - | 5 | 0.75 | 48.78 | 0.78253 | 0.80205 | $w_E$ |
| all | 250 | $w_E$ | 5 | 0.75 | 48.78 | 0.75639 | 0.7659 | $w_E$ |
| Without hill | 250 | $w_E$ | 5 | 0.75 | 48.78 | 0.75639 | 0.7659 | $w_E$ |
| Without coast | 50 | $w_E$ | 5 | 0.75 | 68.29 | 0.74044 | 0.74508 | $w_E$ |
| all | 50 | $w_E$ | 5 | 0.75 | 70.73 | 0.75838 | 0.73115 | $w_E$ |
| Without hill | 50 | $w_E$ | 5 | 0.75 | 70.73 | 0.75838 | 0.73115 | $w_E$ |
| Without coast | 50 | - | 5 | 0.75 | 68.29 | 0.7525 | 0.71854 | $w_E$ |
| all | 50 | - | 5 | 0.75 | 69.51 | 0.77474 | 0.71303 | $w_E$ |
| Without hill | 50 | - | 5 | 0.75 | 69.51 | 0.77474 | 0.71303 | $w_E$ |
| Without coast | 250 | $w_E$ | 5 | 0.75 | 45.12 | 0.72785 | 0.69494 | $w_E$ |
| Without coast | 250 | - | 5 | 0.75 | 45.12 | 0.74555 | 0.69358 | $w_E$ |

**Table A3.** Variable importance ranking of the predictive models.

| Suitability ≥ | Gain | % Area | % Sites |
|---|---|---|---|
| **M1—Expert: Pompeii Study** | | | |
| 0.00 | 0.50 | 49.31 | 100.00 |
| 0.10 | 0.56 | 43.90 | 100.00 |
| 0.20 | 0.63 | 36.58 | 98.78 |
| 0.30 | 0.68 | 29.42 | 91.46 |
| 0.40 | 0.70 | 21.56 | 70.73 |
| 0.50 | 0.72 | 13.17 | 46.34 |
| 0.60 | 0.72 | 6.44 | 23.17 |
| 0.70 | 0.76 | 2.32 | 9.76 |
| 0.75 | 0.78 | 1.31 | 6.10 |
| 0.80 | 0.82 | 0.66 | 3.66 |
| 0.90 | 0.10 | 0.00 | |
| **M2—Expert: Southwest Spain Study** | | | |
| 0.00 | 0.53 | 47.37 | 100.00 |
| 0.10 | 0.57 | 43.31 | 100.00 |
| 0.20 | 0.65 | 34.50 | 98.78 |
| 0.30 | 0.69 | 29.15 | 95.12 |
| 0.40 | 0.74 | 22.97 | 87.80 |
| 0.50 | 0.77 | 17.58 | 76.83 |
| 0.60 | 0.80 | 10.59 | 53.66 |
| 0.70 | 0.82 | 5.08 | 28.05 |
| 0.75 | 0.83 | 3.76 | 21.95 |
| 0.80 | 0.79 | 2.06 | 9.76 |
| 0.90 | 0.91 | 0.22 | 2.44 |
| **M3—Automated: *IQRnorm*** | | | |
| 0.10 | 0.57 | 43.44 | 100.00 |
| 0.20 | 0.64 | 35.10 | 98.78 |
| 0.30 | 0.70 | 28.14 | 95.12 |
| 0.40 | 0.74 | 22.15 | 84.15 |
| 0.50 | 0.77 | 16.00 | 68.29 |
| 0.60 | 0.78 | 9.21 | 42.68 |
| 0.70 | 0.81 | 3.49 | 18.29 |
| 0.75 | 0.84 | 1.76 | 10.98 |
| 0.80 | 0.88 | 0.75 | 6.10 |
| 0.90 | | 0.08 | 0.00 |
| **M4—Automated: $w_E$** | | | |
| 0.10 | 0.59 | 41.49 | 100.00 |
| 0.20 | 0.67 | 33.29 | 100.00 |
| 0.30 | 0.71 | 26.75 | 92.68 |
| 0.40 | 0.74 | 21.52 | 81.71 |
| 0.50 | 0.75 | 16.45 | 67.07 |
| 0.60 | 0.77 | 10.32 | 43.90 |
| 0.70 | 0.76 | 4.04 | 17.07 |
| 0.75 | 0.87 | 1.88 | 14.63 |
| 0.80 | 0.80 | 0.73 | 3.66 |
| 0.90 | | 0.05 | 0.00 |

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
