# Peer review of "Viticulture in the Laetanian Region (Spain) during the Roman Period: Predictive Modelling and Geomatic Analysis"

_geosciences, doi:10.3390/geosciences10060206_

Round 1

Reviewer 1 Report

The focus of the paper is methodological, and as such will be of use to researchers attempting something similar in other case studies and across several scientific fields. The level of replicability is very good, with the scripts used currently available in a github repository.

The method used to identify useful predictors is innovative and nicely executed. However, there was no attempt to quantity the model parsimony, such as the use of information criteria. I think there needs to be an explanation as to why this was not done or was not applicable to this particular modelling approach.

The discussion is good but I think slightly under-sells the value of the author’s modelling approach. As ever, it is the data that don’t fit the model are the most interesting. The revelation that the sites dated to around 25 BC sites are the ones that the model demonstrates how we can learn from modelling and use the results to develop new models that extend to different dimensions, for example temporal ones. I think this could be stated more prominently in the discussion or the conclusions.

The paper is well-written and the figures are most excellent. Here are a few minor suggestions:

56-57 Should read something like ‘This, for example, includes the use of ….’ and remove ‘This means,’ from the start of the next sentence

148 ‘point’ not ‘points’

163 ‘used’ instead of ‘worked’

284-5 ‘air drainage’ ?

Author Response

Review #1

The focus of the paper is methodological, and as such will be of use to researchers attempting something similar in other case studies and across several scientific fields. The level of replicability is very good, with the scripts used currently available in a github repository.                                                

The method used to identify useful predictors is innovative and nicely executed. However, there was no attempt to quantity the model parsimony, such as the use of information criteria. I think there needs to be an explanation as to why this was not done or was not applicable to this particular modelling approach.

Answer: The predictive modeling approach was designed in a way to leave full control to the scientist to decide about the number of independent variables to use. Thus, the scientist has to be aware of the concept of model overfitting. As a quality measure for the model, we decided to use Kvamme’s gain statistic that uses the area percentage of a modeled zone of interest and puts it in relation with the percentage of sites found within that zone of interest. This combines two very important criteria to evaluate predictive models. They have to have both a high accuracy (equivalent to correct predictions) with a high precision (the ability to limit the area of high probability). To reveal model overfitting by using to many useless variables, we applied an external model validation based on a k-fold cross-validation.

The discussion is good but I think slightly under-sells the value of the author’s modelling approach. As ever, it is the data that don’t fit the model are the most interesting. The revelation that the sites dated to around 25 BC sites are the ones that the model demonstrates how we can learn from modelling and use the results to develop new models that extend to different dimensions, for example temporal ones. I think this could be stated more prominently in the discussion or the conclusions.

Answer: You are totally right. Hence, we incorporated this importnat aspect into the conclusion (lines 783-788).

The paper is well-written and the figures are most excellent. Here are a few minor suggestions:

56-57 Should read something like ‘This, for example, includes the use of ….’ and remove ‘This means,’ from the start of the next sentence

Answer: We changed that in the text as suggested.

148 ‘point’ not ‘points’

Answer: We changed that in the text as suggested.

163 ‘used’ instead of ‘worked’

Answer: We changed that in the text as suggested.

284-5 ‘air drainage’ ?

Answer: For a better understanding we changed that sentence into: „Slopes between 5 and 15 % provide sufficient air circulation to reduce the chance of infections with fungal or bacterial diseases. Moreover, erosion and nutrient loss is moderate“.

Reviewer 2 Report

I found this paper incredibly interesting, well written and extremely innovative. I particularly appreciated the clarity of the method and how it was described. The Python code used for this predictive model is proficiently and elegantly structured. Impressive is the webGIS for visualisation. I therefore recommend this publication in its present form, provided a quick sanity check for typos. 

Despite the positive impression I had from this paper, I would like to provide some remarks to the authors about the method. I don't think any of these aspects need to be addressed in the present paper, but they might provide some "food for thought" for future developments of the analytical protocol. I really liked the fact that the authors produced a tailored approach to predictive modelling, and they did not rely on well-established tools based on logistic regression, MaxEnt, neural networks. But in doing so, they were forced to make a series of assumptions.

  • By averaging the value of the covariates in the buffer around the sites (line 159), they assumed on the one side that these values approximate a normal distribution, and on the other side that their frequency inside the buffer is not affected by spatial autocorrelation. Spatial autocorrelation, in turn, might alter the reliability of this statistical parameter, since the central limit theorem assumes independence of the samples used to infer the population statistics.
  • By relying on the 50th to 75th percentile of the normalised IQA, for each covariate, to predict the most suitable areas for wine presses, they implicitly assume that the analysed spatial process is stationary. However, if the relationship between site distribution and covariates is not homogeneous across the study area, and if the spatial variation of the covariates is equally not homogeneous, the assumption is no longer valid. Non-stationarity is suggested by the clustering of the sites (line 442-443) and by the different performance of the model in different sectors of the study area (lines 484-485).

In other words, I believe the authors should address spatial dependence more explicitly in their protocol. Different options can be evaluated, the simplest of which can be spatial decomposition (dividing the study area into sub-areas) or bootstrapping. This would definitely improve the performance of their final predictive model (line 611-612).

Another issue can be the arbitrary selection of the number of covariates (lines 422-423). I believe that the number of covariates should not be established according to an arbitrary threshold, but based on the optimal correlation with predictivity. The solution suggested by the author is to test different models with different variables and thresholds. This is the right approach, but it might be extremely time-consuming if it is done for all the possible permutations (e.g. 4 different models created for 3 variables)! I recommend the development of a multi-model selection method, similar to those used for regression models.

These are just sparse observations, based on my experience, which I think might be helpful for strengthening the future versions of the model. Having said that, I reinforce my appreciation of this methodological approach, its application and its description in this paper.

Author Response

Review #2

I found this paper incredibly interesting, well written and extremely innovative. I particularly appreciated the clarity of the method and how it was described. The Python code used for this predictive model is proficiently and elegantly structured. Impressive is the webGIS for visualisation. I therefore recommend this publication in its present form, provided a quick sanity check for typos.

Answer: Thank you so much for this excellent feedback.

Despite the positive impression I had from this paper, I would like to provide some remarks to the authors about the method. I don't think any of these aspects need to be addressed in the present paper, but they might provide some "food for thought" for future developments of the analytical protocol. I really liked the fact that the authors produced a tailored approach to predictive modelling, and they did not rely on well-established tools based on logistic regression, MaxEnt, neural networks. But in doing so, they were forced to make a series of assumptions.

  • By averaging the value of the covariates in the buffer around the sites (line 159), they assumed on the one side that these values approximate a normal distribution, and on the other side that their frequency inside the buffer is not affected by spatial autocorrelation. Spatial autocorrelation, in turn, might alter the reliability of this statistical parameter, since the central limit theorem assumes independence of the samples used to infer the population statistics.
  • By relying on the 50th to 75th percentile of the normalised IQA, for each covariate, to predict the most suitable areas for wine presses, they implicitly assume that the analysed spatial process is stationary. However, if the relationship between site distribution and covariates is not homogeneous across the study area, and if the spatial variation of the covariates is equally not homogeneous, the assumption is no longer valid. Non-stationarity is suggested by the clustering of the sites (line 442-443) and by the different performance of the model in different sectors of the study area (lines 484-485).

In other words, I believe the authors should address spatial dependence more explicitly in their protocol. Different options can be evaluated, the simplest of which can be spatial decomposition (dividing the study area into sub-areas) or bootstrapping. This would definitely improve the performance of their final predictive model (line 611-612).

Answer: You are right. We were aware of this problem. However, dividing the study area into sub-areas was not possible in our case. Our model is based on 82 archaeological sites. Reducing that number by creating sub-samples would also decrease the quality of the individual predictive models. However, it could be worth to test that aspect.

Another issue can be the arbitrary selection of the number of covariates (lines 422-423). I believe that the number of covariates should not be established according to an arbitrary threshold, but based on the optimal correlation with predictivity. The solution suggested by the author is to test different models with different variables and thresholds. This is the right approach, but it might be extremely time-consuming if it is done for all the possible permutations (e.g. 4 different models created for 3 variables)! I recommend the development of a multi-model selection method, similar to those used for regression models.

These are just sparse observations, based on my experience, which I think might be helpful for strengthening the future versions of the model. Having said that, I reinforce my appreciation of this methodological approach, its application and its description in this paper.

Answer: Thank you very much for those very useful suggestions to refine our predictive modelling approach for future applications. We will definitely keep them in mind.